# Body Image Perception in High School Students: The Relationship with Gender, Weight Status, and Physical Activity

**DOI:** 10.3390/children10010137

**Published:** 2023-01-10

**Authors:** Stefania Toselli, Luciana Zaccagni, Natascia Rinaldo, Mario Mauro, Alessia Grigoletto, Pasqualino Maietta Latessa, Sofia Marini

**Affiliations:** 1Department for Life Quality Studies, University of Bologna, 47921 Rimini, Italy; 2Department of Neuroscience and Rehabilitation, Faculty of Medicine, Pharmacy and Prevention, University of Ferrara, 44121 Ferrara, Italy; 3Department of Biomedical and Neuromotor Sciences, University of Bologna, 40126 Bologna, Italy

**Keywords:** body image, self-perception, physical activity, body shape questionnaire

## Abstract

Body image perception includes body size assessment, body desirability estimation, and perceptions concerning one’s own body shape and size. Adolescence is a period of intense and prompt physical transformation, which changes the perception of one’s body. This represents a critical period for the development of body image. Therefore, the present cross-sectional study aimed to evaluate body image perception and investigate the relationships between it, weight status, sex, and physical activity in a sample of high school students living in Italy. General demographic information and details about physical activity were collected. Body image perception was measured with a body silhouette and two indexes were calculated: the FID (Feel minus Ideal Discrepancy) to evaluate the discrepancy between the perceived current figure and the ideal figure; and the FAI (Feel weight status minus Actual weight status Inconsistency) to observe improper perception of weight status. In addition, body shape concerns were evaluated with the Body Shape Questionnaire (BSQ), in which participants reported the frequency of experiencing negative thoughts about their body shape in the last four weeks. Two hundred and four students were included in the study (155 = female, mean age = 17.13 ± 1.70; 49 = male, mean age = 17.25 ± 1.69). Females felt more concerned about body shape than males (χ^2^ = 11.347, *p* = 0.001). Distinctions emerged in terms of body mass index, the scores of Feel minus Ideal Discrepancy (FID), Feel weight status minus Actual weight status Inconsistency (FAI), the Body Shape Questionnaire (BSQ), and of the silhouette mean comparisons due to sex, weight status, and PA interaction effects (*p* < 0.001). Additionally, 94% of the BSQ variability could be explained by sex, weight status, and PA. Although no direct effects were observed on body image perception, healthy habit promotion, such as physical activity, could positively affect adolescent lives.

## 1. Introduction

Body image is a multidimensional construct, which therefore encompasses perceptual, cognitive, and affective elements concerning one’s own body and the bodies of others [1,2,3]. Body image perception includes body size assessment (how a person perceives his or her body), body attractiveness estimation (what is the type of body that a person considers most attractive), and perceptions related to one’s own body shape and size [4]. Thus, interactions among physiological, cognitive, and sociocultural factors contribute to body image development [5]. During life, people are subjected to physical and psychological changes that affect the perception of their image. Thus, body image is not a static concept but rather a dynamic characteristic influenced by the perceptions of the individual about him- or herself [6]. Adolescence represents a period of intense and prompt physical changes and it therefore also changes the perception of one’s body characteristics, and it can be a critical time for the development of body image [2,6,7,8]. Acceptance of and adaptation to these changes are essential for adolescents’ feelings about their body image [9]. However, these transformations can represent a risk factor in adolescents’ body image perception [6], thereby causing a distorted image of one’s body. Moreover, distorted perception of one’s body image can result from inaccurate evaluations of one’s body, undesirable effects regarding one’s body, miscognition of body-related stimuli, and specific body-related behaviours such as checking of body weight or avoidance of the consideration of body [10]. Worse still, it can result in unhealthy behaviours and adverse psychological outcomes such as an unhealthy diet, low self-esteem, and psychological disturbances [1,11,12].

There are several factors that influence the risk of body image disturbance during adolescence. The body image perception of adolescents has been reported to be associated with bio-social factors such as sex, age, socioeconomic status, and opportunities for health education [13,14]; environmental factors such as media exposure, peers, and school [1,14,15,16,17]; and behavioural factors such as weight control behaviour, physical activity, sexual behaviour, and eating patterns [11,18,19].

Regarding biological factors, the influence of sex on body image is well-known; body image dissatisfaction is more frequent among girls, but it is also present in thin boys [20,21]. 

Behavioural factors play a fundamental role, as they can be modified. Adolescents are responsive to the acquisition of habits and routines that are often influenced by their environment, which can be significant for their future life and of great importance in promoting healthy lifestyles during this time [22]. Among these lifestyle factors, proper nutrition and regular physical activity are of particular importance. Unfortunately, the data in the literature suggest a decline of optimal eating patterns and a decrease in physical activity in adolescents, who frequently assume sedentary behaviour [23].

Furthermore, adolescents are influenced by markets and advertising, which contribute to making their diet unbalanced and excessively caloric; rich in refined fats and sugars; poor in fruit, vegetables, legumes, and fish; and in many cases omitting breakfast [23]. Thus, in 2018, an international study conducted by the WHO found that one in five adolescent (21%) respondents was overweight or obese [24]. In Italy, in 11–15-year-old adolescents, the prevalence of overweight was 16.6% and obesity was 3.2%; excess weight decreased slightly with age and was greater in males (HBSC 2018: https://www.epicentro.iss.it/hbsc/indagine-2018, accessed on 22 February 2022). Distorted perception of weight status is quite prevalent in adolescents [25,26]. High-school-aged students tend to overrate height and underreport weight, thereby decreasing the prevalence of overweight and obesity estimates with lower average self-reported BMI [27]. Proper perception of weight is the key to determining the nutritional habits and weight management of adolescents. Indeed, it has been demonstrated that many overweight students are unlikely to participate in weight control practices [26,28,29,30,31].

As reported above, sports participation and physical activity (PA) can affect body image in adolescents [32,33,34,35]. In general, female and male adolescents who play sports present a lower body dissatisfaction; this can be connected to the role of sports participation in contributing to lower adiposity levels and, consequently, to lower body dissatisfaction in the adolescent population [36,37,38]. However, the associations between body image and sports participation in adolescents are not consistent, especially according to sex. Tebar et al. [39] reported that sports participation was associated with low body dissatisfaction in boys but not in girls. In sports contexts, females and males are not encouraged in the same way due to sex-stereotyped beliefs [40]. This appears in sex disparities in sports participation: in Italy, scant physical activity was higher among females and older adolescents [41]. Morano et al. [40] reported that 41% of girls and 59% of boys 11–19 years old practice sports. Despite the large importance of sports participation, the sport experience of adolescents has not been widely investigated, and the studies on body image concerns in adolescents involved in sports are rare and inconclusive.

Given this information, the main goal of the present study was to evaluate body image perception and to explore the relationships among it, weight status, sex, and physical activity in a sample of high school students living in Italy. Our hypothesis was that there would be a significant interaction effect between weight status, sex, and physical activity on body image perception in the sample. Specifically, we assumed that overweight/obese adolescents with a lower level of reported physical activity would have the lowest body perception and that these effects would be especially evident in females.

## 2. Materials and Methods

A cross-sectional study design was chosen, and data were collected and analysed in a sample of 204 adolescents attending secondary school in Piemonte (northern Italy). All adolescents attending the school could answer the questionnaires. The inclusion criteria were attending the school, completing all the parts of the questionnaires, having parental written consent, and agreeing to participate. The exclusion criteria were not having parental written consent and not completing all the parts of the questionnaires. So, we analysed only the fully completed questionnaires. The study was approved by the Bioethics Committee of the University of Bologna (approval no. 25027).

### 2.1. Outcome Assessment

General information about demographic variables (e.g., sex and age) and physical activity participation, both structured sport (e.g., volleyball with a team and coach) and independent physical activity (e.g., walking, yoga, etc.), was collected by an ad hoc questionnaire that was created for this purpose. The sports participation frequency of each subject was determined by the number of days per week of physical activity during a typical week as declared by the subject. Concerning anthropometric evaluation, participants’ self-reported height and weight were converted to meters and kilograms to calculate body mass index (BMI).

### 2.2. Body Image Perception

The perception of body image was evaluated using the validated Body Silhouette Chart [42,43]. Nine male or female silhouettes, ordered in morphology from emaciation to obesity, were shown to the students, who were asked to select the one which they most desired (ideal figure) as well as the silhouette which they believed was most similar to their own (perceived current figure) [42]. The silhouette series was divided into underweight, normal weight, overweight, and obese [42]. The discrepancy between the perceived current figure and the ideal figure represented the degree of body image dissatisfaction (FID or Feel minus Ideal Discrepancy) [44]. By subtracting the number of the figure selected by the students as the ideal figure from the one selected as their perceived current figure, it was possible to calculate the FID index. A negative FID score indicates that the perceived current figure was thinner than the ideal figure, and a positive score indicates that the perceived current figure was bigger than the ideal figure. An FID score of 0 indicates that the student has chosen the same figure as perceived current and perceived ideal, showing no discrepancy.

The FAI (Feel weight status minus Actual weight status Inconsistency) index was used to evaluated the improper perception of weight status [45]. FAI inconsistency was calculated by subtracting the conventional code assigned to the real weight status of the participant from the code of her or his perceived weight status, which assigns a specific weight status category to each silhouette. The conventional code of real weight status of the participant and the perceived weight were classified according to the BMI assessed by Cole cut-off values by sex and age [46,47] (1 = underweight; 2 = normal weight; 3 = overweight; and 4 = obese). Instead, the classification recommended by Sànchez-Villegas was used for the silhouettes obtained [42]. In this classification method, silhouettes one, two, and three correspond to underweight (=1), silhouettes four and five correspond to normal weight (=2), silhouettes six and seven correspond to overweight (=3), and silhouettes eight and nine correspond to obesity (=4). A positive FAI score indicates that weight status is overestimated, a negative score indicates that weight status is underestimated, and a score of zero indicates no inconsistency in weight status perception.

### 2.3. Body Shape Concerns

Body shape concerns were assessed with a 14-item version [48] of the Body Shape Questionnaire (BSQ) [49], which was validated in Italian by [50]. The BSQ utilizes a six-point Likert scale ranging from one to six (one = never, six = always) to assess how frequently in the past four weeks a participant reported experiencing negative thoughts or feelings about their body shape (e.g., “Have you felt ashamed of your body?”). In addition, the questionnaire investigated if they had tried to control or lose weight with the same six-point Likert scale. Scores from all items are added to get a total score. Higher scores reflected greater body shape concerns. The total score was calculated on the sum of all the values was multiplied by 34/14 and subsequently related to the specific thresholds that refer to the complete form of the questionnaire: a score below 80 indicated “no concern”, a score between 80 and 110 indicated a “slight concern”, a score between 111 and 140 indicated a “moderate concern”, and a score above 140 indicated a “marked concern”.

### 2.4. Statistical Analysis

The statistical analysis was performed with STATA^®^ software, version 17 (Publisher: StataCorp. 2021. Stata Statistical Software: Release 17. College Station, TX, USA, StataCorp LP). Descriptive statistics were calculated and reported such as mean ± standard deviation (SD) for quantitative variables (age, height, weight, BMI, silhouette think to look, silhouette want to be, FID, FAI, and BSQ) and frequency (number of observations) and percentage (%) for qualitative variables (weight status, weight control, body shape concern, and trying to lose weight). The chi-squared test (χ^2^) was used to assess discrepancies in frequencies among categorial and binomial variables. The Shapiro–Wilk test was performed to check the variables’ normal distribution. Due to the presence of predictors with different data levels, an analysis of covariances and the interaction effect between sex, weight status, and physical activity frequency were tested. A stepwise procedure with backward selection was carried out to perform the multiple regression analysis and to assess possible predictors of FID, FAI, and BSQ. Stepwise estimation included only predictors with a significance level for removal from the model of at most *p* < 0.05.

## 3. Results

A total of 204 students, consisting of 155 females (mean age = 17.13 ± 1.70) and 49 males (mean age = 17.25 ± 1.69), were included in the final analysis (Figure 1). Considering weight status, normal weight was the most represented category in both females and males.

Table 1 shows no statistically significant differences in frequencies between sexes except for body shape concern, which displayed more trouble in females. Additionally, male adolescents were less worried about weight loss.

Table 2 shows the mean comparisons and effects of the interaction between sex, weight status, and frequencies of PA. Sex differences were significant only for the desired figure and body shape questionnaire, while all variables related to body perception were influenced by weight status. In the underweight category, FID and FAI scores showed negative discrepancies, indicating a misperception tending to underestimation. Regarding effects of PA frequency, it did not show significant outcomes. However, interaction effects of sex, weight status, and PA frequency were relevant for every variable.

To assess possible predictors of body image dissatisfaction (assessed by FID score), inconsistency of weight status (assessed by FAI score), and body shape concern (assessed by BSQ score), three backward multiple regressions were performed. Table 3 shows the predictive model results. Regarding FID, weight and all the categories of PA frequency were positive predictors, whereas the conditions of being younger, male, and having tried to lose weight sometimes, were found to be associated with better body image satisfaction. Although the whole model explained 60% of the FID variance, adolescent weight seemed to be the most influential variable (*η*^2^ = 0.417).

Concerning FAI, being underweight, normal weight, and never trying to lose weight appeared as negative predictors of weight status estimation. Contrarily, body weight overestimation was associated with all the categories of PA frequency. The model explained 24% of the variance and had a highly significant R^2^. Being male and not obese was significantly associated with a lower score of body shape concern. The BSQ score increased as body image dissatisfaction and PA frequency increased. The model explained 94% of the variance.

Finally, Figure 2, Figure 3 and Figure 4 shows the linear prediction of the BSQ influenced by PA frequency categories and the weight status (A), PA frequency categories, FID values (B), and PA frequency categories and sex (C). Lower BSQ values corresponded to 4–5 days per week of PA in each plot except for overweight subjects (Figure 2, Figure 3 and Figure 4, green line).

## 4. Discussion

The present study aimed to evaluate body image perception in a sample of high school students living in Italy and to explore the relationships between it, sex, weight status, and physical activity.

In the present study, no gender differences were observed in terms of weight status and weight control. Nonetheless, girls showed higher values of weight concerns. In addition, this study confirmed the presence of sex differences in body image perception. Admittedly, the absence of differences in weight status was unexpected since in Italy, a significantly higher proportion of males were overweight/obese compared with females [51,52]. These data confirmed that girls in this age range present significant weight concerns, as seen in similar studies [53,54,55]. It is noteworthy that even though the present study showed a greater prevalence of weight concern among girls, males also reported it. This result is of particular interest, and it cannot be neglected given the established sex-specific patterns, such as muscularity-oriented disordered eating [54,56]. Relatedly, a longitudinal study by Bucchianeri et al. [57] showed that in adolescence, body dissatisfaction increased with time in both sexes, but the levels of body dissatisfaction were remarkably higher in girls. Generally, body image dissatisfaction and weight control behaviour are greater issues for girls compared with males [55]. In our study, FID was higher in females, although it was not a significant difference between the sexes. Girls were found to be more inclined than boys to emphasize the aesthetic values of their bodies rather than functional ones. Moreover, females reported more dissatisfaction with both values than boys [58]. Consistent with this, even in childhood, girls seemed to be more conscious about how their body weight affects their appearance compared with boys [59]. The difference between real and ideal silhouettes with a higher incidence in girls may result in a differential diagnosis of body dysmorphic disorder, since it is more prevalent in women, beginning at the age of between 15 and 30 years [55]. The beauty standard currently imposed by media that values thinness reinforces this condition, which may lead to the adoption of restrictive diets regardless of the real need and may contribute to the occurrence and maintenance of low weight as well as the compromise of eating habits [60] and the presence of mental disorders [61]. For this purpose, it is noteworthy that even though the weight control outcome did not show significant differences between sexes, the percentage of individuals who reported putting attention on weight control is high in both sexes (76.77% in females and 69.39% in males), although a higher percentage of females declared that they tried to lose weight often or always compared to males (38.06% vs. 16.66%). Undoubtedly, the role of gender in body image has been changing; a study by Dzielska et al. [62] targeted adolescents from 26 countries from 2001/2002 to 2017/2018 and reported that the overall age-adjusted prevalence rates of weight-reduction behaviours (WRB) were 10.2% among boys and 18.0% among girls. The prevalence was higher for girls, but in more recent surveys, sex differences in WRB decreased, and a significant increase in the percentage of WRB among boys was observed in most countries. Therefore, from the present results of our study, being a male seems to be still a protective factor. Additionally, being male and having sometimes tried to lose weight was found to be associated with better body image satisfaction. Likewise, being male and not obese was significantly associated with a lower score of body shape concern.

Regarding the second investigated factor, weight status, overweight and obese students showed higher concerns, while no concern and slight concern were observed in underweight and normal weight subjects. FID was found to be related to a growing dissatisfaction with body image or to an increase in weight status. Thus, being underweight is the only category whose members would desire to be fatter, whereas in the other categories, the desire to be thinner prevailed. A relationship with weight also emerged from the regression analysis since weight was significantly associated with body image dissatisfaction. Relatedly, de Pinho et al. [55] reported a significant association between nutritional status and body image in adolescents; in particular, underweight and overweight adolescents were dissatisfied with their body image. Admittedly, weight status is associated with body dissatisfaction, weight concerns, and dietary restraints [63], potentially resulting in both restrictive and binge-purge disordered eating [64,65]. For this reason, studies regarding body image perception in this age group may be valuable for the early detection of disorders. Considering FAI, overweight subjects overestimate their weight status more than normal weight and obese subjects, while a misperception tending to underestimation is observed in underweight students. In addition, the regression analysis revealed that being underweight, normal weight, and having never trying to lose weight are negative predictors, and are thus protective factors, of weight status estimation. The number of teenagers who perceive themselves as overweight and adopt unhealthy strategies to lose weight is constantly increasing, as was previously reported by epidemiological surveys carried out in Italy [66].

The frequency of PA did not show significant associations with body image perception. The influence of sport participation and physical activity (PA) on body image in adolescents is not consistent. In general, adolescents of both sexes who play sports present a lower body dissatisfaction; however, associations between body image and sports practice in adolescents are not consistent, especially according to sex [32,33,34,35,36,37,38,39]. Our study showed that only intense physical activity effected perception [21]. From the results of the regression analysis, PA was associated with FID, FAI, and BSQ. In particular, all the categories of PA frequency were significantly associated with body image dissatisfaction and with body weight overestimation. Similarly, being physically active seemed to be associated with body shape concerns. Arguably, this is related to wider body awareness, which might contribute to generating more concern about body shape.

From the results of the present study, sex and weight status were found to be the prevailing factors influencing body image perception, while physical activity seemed to have a less decisive influence. Therefore, interventions should be directed at factors that influence the acceptance of their own body and the promotion of a healthy lifestyle. The sex-specific patterns in body image investigated in the present study underlined the need to focus on the risk factors for body image misperception and concerns by sex and weight status categories. Moreover, specific health promotion initiatives taking these aspects into account should be implemented.

The primary limitation of the study was the cross-sectional nature of the data, which limited causal inferences. A longitudinal study design during the whole adolescent period would be valuable to determine the direction of the relationship among the variables considered. In addition, height and weight were self-reported due to the COVID-19 public health emergency period, and this could have influenced the accuracy of the results. Information about sociodemographic variables was not collected, so it was not possible to evaluate the influence of these factors in explaining some of the possible links between physical activity participation and body perception. The sample size was limited, and further research with a larger sample is needed to verify the validity of the present results.

## 5. Conclusions

Although the prevalence of high school students who are of normal weight is high, many adolescents exhibit an altered body image and body shape perception, especially in the female sex. The daily practice of physical activity should reduce concerns related to body shape and facilitate better self-awareness. Despite more evidence being needed, the promotion of healthy habits such as physical activity could be an optimal strategy to improve mental wellness in adolescents.

## Figures and Tables

**Figure 1 children-10-00137-f001:**
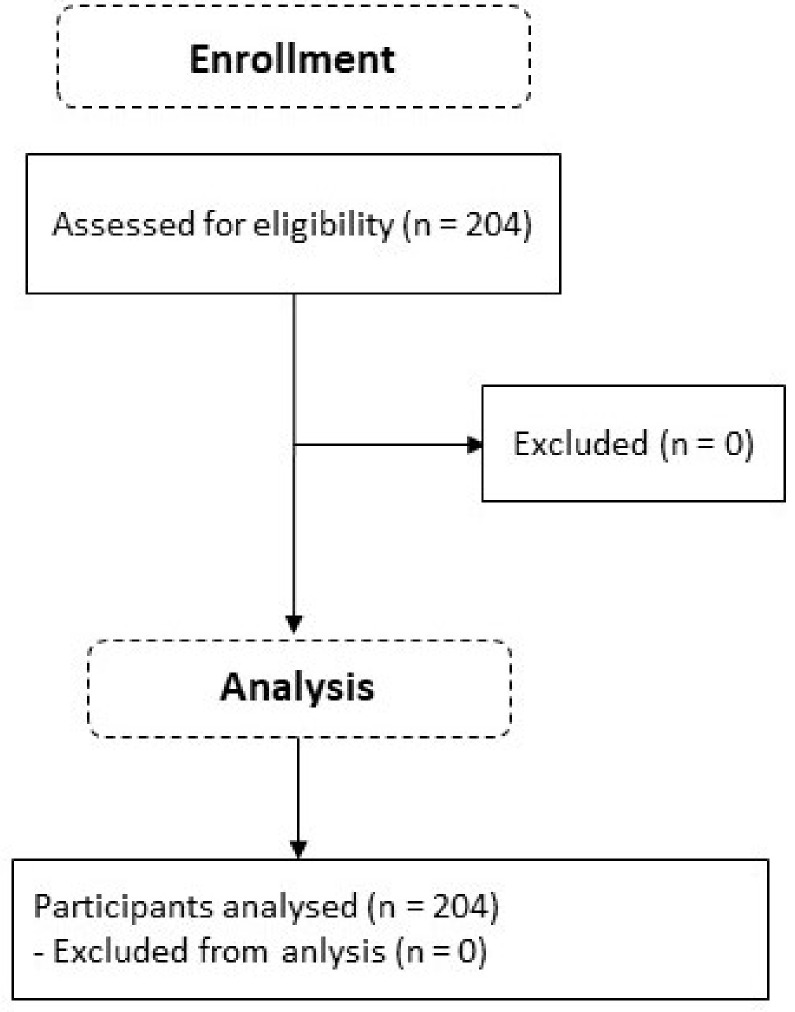
Sample flowchart.

**Figure 2 children-10-00137-f002:**
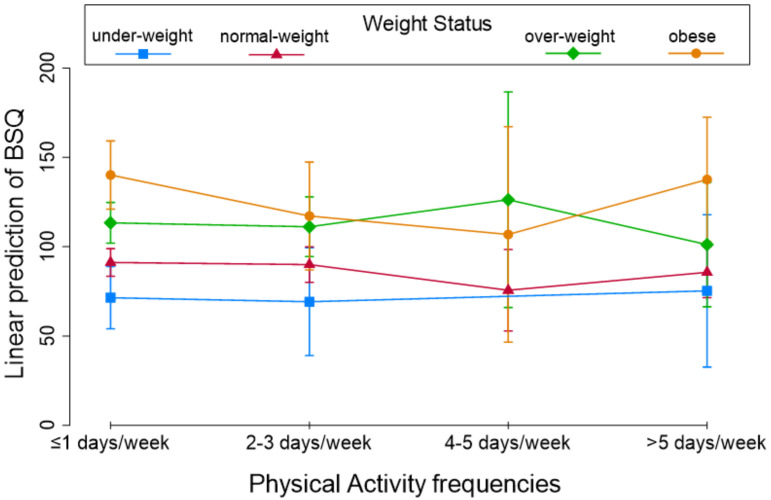
Linear prediction of BSQ by Weight Status for four Physical Activities frequencies.

**Figure 3 children-10-00137-f003:**
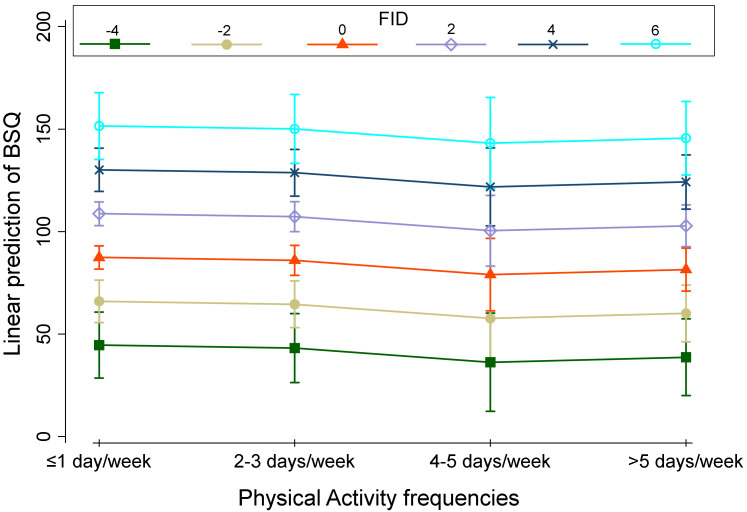
Linear prediction of BSQ by FID for four Physical Activities frequencies.

**Figure 4 children-10-00137-f004:**
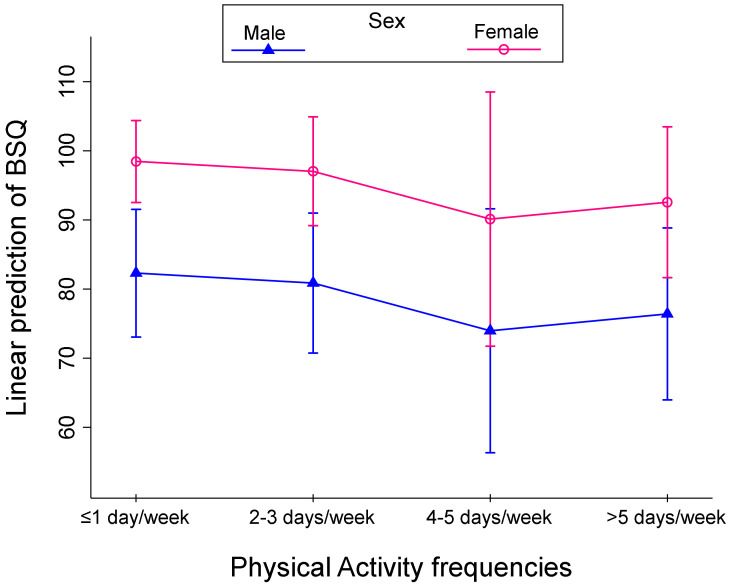
Linear prediction of BSQ by sex for four Physical Activities frequencies.

**Table 1 children-10-00137-t001:** Frequencies of weight status, body shape concern, weight control and trying to lose weight by sex.

	Sex [n (%)]		Statistics	
	F = 155 (75.98)	M = 49 (24.02)	χ^2^	*p*
Weight Status			2.945	0.400
Underweight	14 (9.03)	4 (8.16)		
Normal weight	97 (62.58)	26 (53.06)		
Overweight	33 (21.29)	12 (24.49)		
Obese	11 (7.10)	7 (14.29)		
Body shape concern (BSQ)			11.347	0.010
No	56 (36.13)	26 (53.06)		
Some	41 (26.45)	17 (34.69)		
Moderate	29 (18.71)	4 (8.16)		
Marked	29 (18.71)	2 (4.08)		
Weight control			1.083	0.298
No	36 (23.23)	15 (30.61)		
Yes	119 (76.77)	34 (69.39)		
Tried to lose weight			4.293	0.231
Never	42 (27.1)	16 (33.33)		
Sometimes	54 (34.84)	21 (43.75)		
Often	45 (29.03)	7 (8.33)		
Always	14 (9.03)	4 (8.33)		

Note: F, female; M, male; χ2, chi-squared test; *p*, *p*-value; %, percentage; N, number of observations.

**Table 2 children-10-00137-t002:** Mean comparisons and interaction effects between sex, weight status, and physical activity frequency.

Variable	Sex					Weight Status					Physical Activity Frequency						Interaction Effect			
	Male	Female	Stats		Under	Normal	Over	Obese	Stats		≤1 d·w^−1^	2–3 d·w^−1^	4–5 d·w^−1^	>5 d·w^−1^	Stats		Sex # WS # PA Frequencies			
	Mean (±SD)	Mean (±SD)	F _(1, 202)_	*p*	Mean (±SD)	Mean (±SD)	Mean (±SD)	Mean (±SD)	F _(3, 200)_	*p*	Mean (±SD)	Mean (±SD)	Mean (±SD)	Mean (±SD)	F _(3, 200)_	*p*	F _(25, 178)_	*p*	adj-R^2^	η^2^
Age	17.25 (1.69)	17.13 (1.70)	0.18	0.67	17.85 (1.52)	16.99 (1.74)	17.48 (1.56)	16.83 (1.62)	2.20	0.09	17.14 (1.67)	17.04 (1.72)	17.76 (1.82)	17.28 (1.72)	0.51	0.67	1.17	.27	0.02	0.14
Height	174.96 (7.59)	163.57 (6.72)	100.16	ƚ	48.44 (3.78)	57.78 (8.00)	74.99 (10.8)	87.56 (13.79)	0.80	0.49	165.59 (8.17)	166.03 (8.88)	167.67 (9.86)	169.58 (7.94)	1.67	0.18	4.71	ƚ	0.31	0.40
Weight	73.00 (16.86)	60.34 (11.75)	34.56	ƚ	166.5 (6.69)	165.62 (8.26)	167.6 (9.43)	167.82 (9.11)	99.34	ƚ	63.20 (14.52)	62.10 (14.30)	67.62 (13.36)	65.53 (12.96)	0.63	0.60	18.41	ƚ	0.68	0.72
BMI	23.75 (4.82)	22.52 (3.95)	3.23	0.07	17.46 (0.70)	20.99 (1.79)	26.58 (1.85)	31.71 (2.58)	293.07	ƚ	22.96 (4.29)	22.36 (4.03)	24.12 (4.79)	22.75 (4.09)	0.56	0.64	37.92	ƚ	0.82	0.84
STL	4.27 (1.66)	4.06 (1.55)	0.64	0.42	2.50 (0.86)	3.50 (1.08)	5.78 (1.01)	6.17 (1.50)	73.98	ƚ	4.17 (1.63)	4.10 (1.52)	4.22 (1.56)	3.81 (1.50)	0.39	0.76	9.34	ƚ	0.51	0.57
SWB	3.51 (1.02)	2.97 (1.01)	10.38	§	2.72 (0.75)	2.89 (1.01)	3.64 (0.96)	3.61 (0.98)	9.08	ƚ	3.17 (1.03)	3.14 (1.10)	3.11 (0.78)	2.73 (0.96)	1.30	0.28	2.30	ƚ	0.14	0.24
FID	0.76 (1.56)	1.08 (1.41)	1.92	0.17	−0.22 (0.73)	0.62 (1.28)	1.93 (0.91)	2.56 (1.76)	28.41	ƚ	1.00 (1.41)	0.97 (1.61)	1.11 (1.54)	1.08 (1.29)	0.05	0.99	4.02	ƚ	0.27	0.36
FAI	0.14 (0.50)	0.14 (0.51)	0.01	0.93	−0.50 (0.51)	0.12 (0.42)	0.38 (0.49)	0.28 (0.57)	16.40	ƚ	0.14 (0.55)	0.12 (0.42)	0.11 (0.33)	0.15 (0.54)	0.04	0.99	2.49	ƚ	0.16	0.26
BSQ	82.17 (27.78)	101.03 (34.55)	12.10	ƚ	71.37 (11.24)	89.13 (29.52)	112.2 (35.20)	132.76 (32.86)	19.42	ƚ	99.05 (35.84)	95.17 (31.17)	84.73 (29.53)	92.66 (33.49)	0.71	0.55	3.54	ƚ	0.24	0.33

Note: BMI, body mass index; STL, silhouette think to look; SWB, silhouette want to be; FID, Feel minus Ideal Discrepancy; FAI, Feel weight status minus Actual weight status Inconsistency; BSQ, Body Shape Questionnaire; SD, standard deviation; WS, weight status; PA, physical activity; F, statistic test of Snedecor–Fisher; d, days; w, week; *p*, *p*-value; adj-R^2^, adjusted R-squared; η^2^, eta-squared effect size; #, interaction; §, *p* < 0.01; ƚ, *p* < 0.001.

**Table 3 children-10-00137-t003:** Predictors of FID, FAI, and BSQ: multiple regression analysis results.

FID (N = 202)						
Source	SS	df	MS	F _(9, 193)_	p	adj-R^2^
Model	388.62	9	43.18	35.26	<0.001	0.604
Residual	236.39	193	1.22			
Total	625	202	3.09			
Variable		β	SE	*t*	*p*	*η^2^*
Height		−0.633	0.013	−4.89	<0.001	0.11
Weight		0.795	0.007	11.75	<0.001	0.417
Age		−0.12	0.047	−2.58	0.01	0.033
PA frequency						0.092
≤1 d·w^−1^		0.878	0.208	4.22	<0.001	
2–3 d·w−^1^		0.887	0.208	4.26	<0.001	
4–5 d·w^−1^		0.911	0.208	4.38	<0.001	
>5 d·w^−1^		0.893	0.213	4.2	<0.001	
Sex (male)		−0.619	0.231	−2.68	<0.01	0.036
TLW (sometimes)		−0.315	0.166	−1.91	0.05	0.019
FAI (N = 203)						
Source	SS	df	MS	F _(7, 196)_	p	adj-R^2^
Model	15.17	7	2.17	10.4	<0.001	0.24
Residual	40.83	196	0.21			
Total	53	203	0.28			
Variable		β	SE	t	p	η^2^
Weight Status						0.202
Underweight		−0.858	0.122	−7.01	<0.001	
Normal weight		−0.239	0.072	−3.34	0.001	
PA frequency						0.178
≤1 d·w^−1^		0.415	0.068	6.13	<0.001	
2–3 d·w−^1^		0.376	0.081	4.63	<0.001	
4–5 d·w^−1^		0.314	0.162	1.93	0.05	
>5 d·w^−1^		0.438	0.108	4.06	<0.001	
TLW (never)		−0.152	0.071	−2.13	<0.05	0.022
BSQ (N = 202)						
Source	SS	df	MS	F _(9, 193)_	p	adj-R^2^
Model	1997339	9	221926.56	344.42	<0.001	0.939
Residual	124357.5	193	644.34			
Total	2121696.4	202	10503.45			
Variable		β	SE	t	p	η^2^
Sex (male)		−16.159	4.45	−3.63	<0.001	0.064
FID		10.688	1.565	6.83	<0.001	0.194
Weight Status						0.081
Underweight		−38.557	10.059	−3.83	<0.001	
Normal weight		−29.164	7.732	−3.77	<0.001	
Overweight		−20.033	7.593	−2.64	0.01	
PA frequency						0.516
≤1 d·w^−1^		116.868	8.218	14.22	<0.001	
2–3 d·w−^1^		115.445	8.752	13.19	<0.001	
4–5 d·w^−1^		108.536	12.186	8.91	<0.001	
>5 d·w^−1^		110.967	9.377	11.83	<0.001	

Note: SS, sum of squares; MS, mean of squares; df, degree of freedom; *F*, Snedecor–Fisher’s statistical test; β, regression coefficient; SE, standard error; *t*, student’s statistic test *p*, *p*-value; adj-R^2^, adjusted R^2^; η^2^, eta-squared.

## Data Availability

Not applicable.

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
