# Peer review of "Body Image Perception in High School Students: The Relationship with Gender, Weight Status, and Physical Activity"

_children, 2023, doi:10.3390/children10010137_

Round 1

Reviewer 1 Report

Thank you for the opportunity to review this manuscript. It reports on an important topic of body image but I believe the manuscript in its current form requires major revision before it would be ready for publication. Additional information is needed about included variables and the discussion should be revised to avoid over-interpretation of results.

Abstract:

·       Please be sure to define all acronyms the first time they are used in the abstract.

·       There are many variables/acronyms included in the results that are not clearly linked to variables in the methods.

·       The final line of the abstract refers to “healthy habit promotion” but it is not clear from the abstract what this is referring to. Is this physical activity practice?

Introduction:

·       Ln 29 introduces the terms body size assessment and body attractiveness estimation but these need to be clearly defined for the reader. Similarly, Ln 42 refers to “checking or avoidance” but more information is needed to clarify what these behaviors entail.

Methods:

·       Ln 103: What is meant by “an ad hoc questionnaire?”

·       The terms “sport” and “physical activity” practice seem to be used interchangeably but are not clearly defined. Did the survey ask only about structured sports (e.g., football with a team and coach) or was independent physical activity (e.g., walking, yoga, weight lifting solo) also included. This has an enormous impact on the interpretation of the data.

·       The variables of “weight control” and “tried to lose weight” are mentioned in the Results but are not described in the Methods. How were these data gathered?

·       Ln. 160-162: It is not necessary to list all the statistical components of ANOVA testing that are in the table.

Results:

·       Ln 172-173 seem to contain formatting instructions that should not be included in the final manuscript

·       How were the various graphs for Figure 2 selected?  Given that PA Frequency is included in all three graphs, it might be helpful to be consistent and have this always be the X-axis (as in B and C, but not in A).

Discussion:

·       Ln 217: What is meant by “a remarkable percentage?” This is a vague descriptor.

·       Ln 224-225: This statement needs to be supported by a reference or made clear that it is still referring to Ref. 55.

·       Ln 241-245 describe changes over time. Include the years in which these data were gathered to provide context to this comparison since readers will not see the title of the referred-to article in the text of this discussion.

·       Ln 273 states that BSQ and FAI were better for students practicing PA 4-5 days per week compared to less active youth, but this comparison was not statistically significant. If this result is discussed in this section, it must include reference to the lack of statistical significance to avoid over-interpretation.

·       Please include discussion of limitations of the current study, specifically the cross-sectional nature of the data. It is impossible to determine the direction of the relationship among these variables and is feasible that youth who are dissatisfied with their body image would be more active in an effort to lose weight.

Throughout:

·       Please review for English grammar, gender-inclusive language and person-first language.

Author Response

Answers to reviewer.

Thank you for the opportunity to review this manuscript. It reports on an important topic of body image but I believe the manuscript in its current form requires major revision before it would be ready for publication. Additional information is needed about included variables and the discussion should be revised to avoid over-interpretation of results.

Answer (A): Thank you so much for your useful suggestions and comments to improve our study. We carefully read all of your points and below you can see our answers and improvements.

Abstract:

  • Please be sure to define all acronyms the first time they are used in the abstract.

A: We defined all the acronyms in the abstract.

  • There are many variables/acronyms included in the results that are not clearly linked to variables in the methods.

A: We defined all the acronyms in the abstract.

  • The final line of the abstract refers to “healthy habit promotion” but it is not clear from the abstract what this is referring to. Is this physical activity practice?

A: The phrase was fixed.

Introduction:

  • Ln 29 introduces the terms body size assessment and body attractiveness estimation but these need to be clearly defined for the reader. Similarly, Ln 42 refers to “checking or avoidance” but more information is needed to clarify what these behaviors entail.

A: The terms were better explained.

Methods:

  • Ln 103: What is meant by “an ad hoc questionnaire?”

A: A questionnaire created for a specific purpose without any plan for repetition. We added a sentence to better specify this concept.

  • The terms “sport” and “physical activity” practice seem to be used interchangeably but are not clearly defined. Did the survey ask only about structured sports (e.g., football with a team and coach) or was independent physical activity (e.g., walking, yoga, weight lifting solo) also included. This has an enormous impact on the interpretation of the data.

A: We asked for any kind of sport or physical activity, so both structured sport and independent physical activity.

  • The variables of “weight control” and “tried to lose weight” are mentioned in the Results but are not described in the Methods. How were these data gathered?

A: This information was part of the questionnaire; we add this part in the description of the questionnaire (lines 154-155).

  • Ln. 160-162: It is not necessary to list all the statistical components of ANOVA testing that are in the table.

A: Thank you for the suggestions, we removed this part.

Results:

  • Ln 172-173 seem to contain formatting instructions that should not be included in the final manuscript

A: Sorry for the inconvenient, we removed the two lines.

  • How were the various graphs for Figure 2 selected?  Given that PA Frequency is included in all three graphs, it might be helpful to be consistent and have this always be the X-axis (as in B and C, but not in A).

A: We improved figure 2 following your suggestion.

Discussion:

  • Ln 217: What is meant by “a remarkable percentage?” This is a vague descriptor.

A: We corrected the sentence.

  • Ln 224-225: This statement needs to be supported by a reference or made clear that it is still referring to Ref. 55.

A: We added the refence and completed the sentence.

  • Ln 241-245 describe changes over time. Include the years in which these data were gathered to provide context to this comparison since readers will not see the title of the referred-to article in the text of this discussion.

A: Thank you for the suggestions, the years were added

  • Ln 273 states that BSQ and FAI were better for students practicing PA 4-5 days per week compared to less active youth, but this comparison was not statistically significant. If this result is discussed in this section, it must include reference to the lack of statistical significance to avoid over-interpretation.

A: Thank you for the suggestions, the references were added, and the sentence modified.

  • Please include discussion of limitations of the current study, specifically the cross-sectional nature of the data. It is impossible to determine the direction of the relationship among these variables and is feasible that youth who are dissatisfied with their body image would be more active in an effort to lose weight.

A: The limitations of the current study were added in lines 315-323.

Throughout:

  • Please review for English grammar, gender-inclusive language and person-first language.

A: Thank you for the suggestion, we review the English grammar and fixed the gender-inclusive and person-first language.

Reviewer 2 Report

Title: Body image perception in high school students: the effects of gender, weight status and physical activity practice

Article Type: Article

Summary

In this study, the authors evaluated body image perception and examined the relationship between it, weight status, sex, and physical activity in 204 students living in Italy. The results indicated that females had more worried about their body rather than males. The results also showed difference for dependent variables between groups.

 Points

 Please add mean age for participants (SD) to the abstract

Please talk more about research method, you didn’t describe anything about method in the abstract, for example, the type of research, sampling method, did you use from scale for, what scales? etc.

The results in abstract are vague, you should clarify it and present it better.

Please add a conclusion to the abstract.

L 18, you shouldn’t start the sentence with number, please write “two hundred and four” instead “204”.

I think the purpose of the research mentioned in the abstract does not match the title properly, so please write it more clearly.

L 34, why “himself”, were their participants male?

L46, L67, there is an extra space.

How did you select the participants? What was the sampling methos? Why did you select only 204 participants? How you did calculate sample size? Please clarify this in the method section.

L 103, you wrote, “The sports practice frequency of each subject was determined by the number of days per week of physical activity practice during a typical week as declared by the subject” why did you use from standard questionnaire like The International Physical Activity Questionnaires (IPAQ) for measuring PA?

Please add inclusion and exclusion criteria to the method section.

 Does the tools you mentioned (Body image perception and Body shape concerns), are suitable for adolescents? Please clarify it in the manuscript? please add it reliability and validity of these tools for using in adolescents.  

L154, “The Shapiro–Wilk test was performed to check variables’ normality distribution.” Shapiro–Wilk test is for sample lower than 50, because of your sample, you should report Kolmogorov–Smirnov test instead of Shapiro–Wilk.

Why did you use “Analysis of Covariances” for you study? Please clarify it.

The quality of Figure 2 is low.

L285, you wrote “In conclusion”, and in the next paragraph you talk again about conclusion, please clarify it.

Please talk more about limitation of the study and even application of the results.

Please use more recent reference regarding your topic, for example, Gualdi-Russo E, Rinaldo N, Zaccagni L. Physical Activity and Body Image Perception in Adolescents: A Systematic Review. International Journal of Environmental Research and Public Health. 2022 Oct 13;19(20):13190.

Author Response

Authors answers.

Title: Body image perception in high school students: the effects of gender, weight status and physical activity practice

Article Type: Article

Summary

In this study, the authors evaluated body image perception and examined the relationship between it, weight status, sex, and physical activity in 204 students living in Italy. The results indicated that females had more worried about their body rather than males. The results also showed difference for dependent variables between groups.

Answer (A): We would like to thank the reviewer to all the useful comments and suggestions to improve our manuscript. We carefully read all your points and below you can see our answers and improvements.

 Points 

 Please add mean age for participants (SD) to the abstract

A: We added the mean age in the abstract.

Please talk more about research method, you didn’t describe anything about method in the abstract, for example, the type of research, sampling method, did you use from scale for, what scales? etc.

The results in abstract are vague, you should clarify it and present it better.

Please add a conclusion to the abstract.

L 18, you shouldn’t start the sentence with number, please write “two hundred and four” instead “204”.

I think the purpose of the research mentioned in the abstract does not match the title properly, so please write it more clearly.

A: We fixed the title according to the aim of the study. We corrected the abstract according to this suggestion, taking into consideration the Journal guidelines about the limited number of words for the abstract (200).

L 34, why “himself”, were their participants male?

A: We fixed the phrase

L46, L67, there is an extra space.

A: We removed the extra space.

How did you select the participants? What was the sampling methos? Why did you select only 204 participants? How you did calculate sample size? Please clarify this in the method section.

A: We proposed to participate to all the student of the school, but only 204 decided to complete the questionnaire. We add this explanation in the method section.

L 103, you wrote, “The sports practice frequency of each subject was determined by the number of days per week of physical activity practice during a typical week as declared by the subject” why did you use from standard questionnaire like The International Physical Activity Questionnaires (IPAQ) for measuring PA?

A: Originally, we thought about the possibility to use the IPAQ for measuring PA, but after speaking with the teachers of the school we decided to not use too much questionnaire, because it might discourage the students from carefully filling in the questionnaires. In addition, we preferred to have not too much confounding variables.

Please add inclusion and exclusion criteria to the method section.

A: The inclusion and exclusion criteria were added in lines 106-112.

 Does the tools you mentioned (Body image perception and Body shape concerns), are suitable for adolescents? Please clarify it in the manuscript? please add it reliability and validity of these tools for using in adolescents.  

A: Thank you for the suggestion. We added a sentence to clarify this aspect.

L154, “The Shapiro–Wilk test was performed to check variables’ normality distribution.” Shapiro–Wilk test is for sample lower than 50, because of your sample, you should report Kolmogorov–Smirnov test instead of Shapiro–Wilk.

A: Despite following the Weak Law of Large numbers and the Central Limit Theorem, in any random variables iid (independent and identically distributed) with the  converges in probability to  when nàinfinite and  (n>50 for health sciences; Statistical Inference, Casella and Berger), the assumption of finite variance is essentially necessary for convergene to normality, and we have no automatic way ofknowing how good the approximation is. The goodness of the approximation is a fuction of the original distribution, and so must be checked case by case. It can be always use for a first rough calculation. In order to assess a more precise distribution description, the Kolmogorov-Smirnov test if less powerful than Shapiro-Wilk for testing normality (M. A. Stephens (1974) EDF Statistics for Goodness of Fit and Some Comparisons, Journal of the American Statistical Association, 69:347, 730-737).

Why did you use “Analysis of Covariances” for you study? Please clarify it.

A: We preferred to use the ANCOVA because it allows to be independent variables of any data level, as previously suggested (Joann G. Elmore MD, MPH, in Jekel's Epidemiology, Biostatistics, Preventive Medicine, and Public Health, 2020).

The quality of Figure 2 is low.

A: We improved figure 2.

L285, you wrote “In conclusion”, and in the next paragraph you talk again about conclusion, please clarify it.

Please talk more about limitation of the study and even application of the results.

A: We phrase was fixed.

Please use more recent reference regarding your topic, for example, Gualdi-Russo E, Rinaldo N, Zaccagni L. Physical Activity and Body Image Perception in Adolescents: A Systematic Review. International Journal of Environmental Research and Public Health. 2022 Oct 13;19(20):13190.

A: Thank you for the suggestion, we added the article to the manuscript and for future research we will consider this article

Round 2

Reviewer 1 Report

I appreciate the author’s efforts to revise the manuscript. However, I still have concerns that review comments were not adequately addressed in the updated manuscript. I have three main concerns: 1) The abstract does not provide enough detail of the methods of the study. Instead, it jumps from purpose to results without providing description of variables or analyses used in the study. 2) Significant revision for English language and style is still needed. Many of the revisions have actually made the manuscript more difficult to read. 3) The explanation given in the author response to comments about whether the physical activity variable collected pertained to all physical activity or just organized sport was not added to the manuscript. This is a vital part of how results are interpreted and must be made clear to the reader.

Additional minor comments:

-Ln 53 begins with the term “biological factors” but goes on to discuss many non-biological factors. Consider re-writing this section.

-Ln 154-155: Were the weight control and lose weight questions answered with the same 1-6 Likert scale? Please specify

-Ln 293-296: My initial concern is still not addressed. This paragraph is discussing findings in the current study that did not reach statistical significance and that must be clearly stated. It is generally considered inappropriate to discuss non-significant findings in this way as it is misleading to the reader.

Author Response

I appreciate the author’s efforts to revise the manuscript. However, I still have concerns that review comments were not adequately addressed in the updated manuscript.

Authors (A): Thank you so much for your useful suggestions and comments to improve our study. We carefully read all of your points and below you can see our answers and improvements.

I have three main concerns:

1) The abstract does not provide enough detail of the methods of the study. Instead, it jumps from purpose to results without providing description of variables or analyses used in the study.

A: We add the detail of the methods of the study in the abstract (lines 19-26). We hope that now the abstract could be clearer.

2) Significant revision for English language and style is still needed. Many of the revisions have actually made the manuscript more difficult to read.

A: Thank you for your suggestion, we tried to review the English again.

3) The explanation given in the author response to comments about whether the physical activity variable collected pertained to all physical activity or just organized sport was not added to the manuscript. This is a vital part of how results are interpreted and must be made clear to the reader.

A: The part about physical activity was added in the manuscript (lines 122-123).

Additional minor comments:

-Ln 53 begins with the term “biological factors” but goes on to discuss many non-biological factors. Consider re-writing this section.

A: The world “biological” was eliminated.

-Ln 154-155: Were the weight control and lose weight questions answered with the same 1-6 Likert scale? Please specify

A: We fixed the explanation in lines 163-164.

-Ln 293-296: My initial concern is still not addressed. This paragraph is discussing findings in the current study that did not reach statistical significance and that must be clearly stated. It is generally considered inappropriate to discuss non-significant findings in this way as it is misleading to the reader.

A: Sorry for the inconvenient, we removed the phrase.

Reviewer 2 Report

Thanks to the corrections made by the dear authors, however, as I mentioned in the previous review , unfortunately this manuscript cannot add anything to the research literature and similar research has been done in the past.

Author Response

We are sorry to read about it, we tried to follow all the reviewers suggestions ans to improve our article.